# Factors Associated with Mortality and Short-Term Patient Outcomes for Hip Fracture Repair in the Elderly Based on Preoperative Anticoagulation Status

**DOI:** 10.3390/geriatrics10020054

**Published:** 2025-04-04

**Authors:** Vimal Desai, Priscilla H. Chan, Kathryn E. Royse, Ronald A. Navarro, Glenn R. Diekmann, Kent T. Yamaguchi, Elizabeth W. Paxton, Chunyuan Qiu

**Affiliations:** 1Department of Anesthesiology, Kaiser Permanente, Baldwin Park, CA 91706, USA; vimal.desai@kp.org (V.D.); chunyuan.x.qiu@kp.org (C.Q.); 2Medical Device Surveillance & Assessment, Kaiser Permanente, San Diego, CA 92108, USAliz.w.paxton@kp.org (E.W.P.); 3Department of Orthopedics, Kaiser Permanente, Harbor City, CA 90710, USA; 4Department of Orthopedics, Kaiser Permanente, Baldwin Park, CA 91706, USA; 5Department of Orthopedics, Kaiser Permanente, Santa Rosa, CA 95403, USA

**Keywords:** hip fracture repair, geriatric, mortality, risk factors, anticoagulants, cohort study

## Abstract

**Background:** The one-year mortality risk for elderly patients undergoing proximal femur fracture repair surgery is three to four times higher compared to the general population. Other than time to surgery, risk factors for postoperative morbidity and mortality following surgery are poorly understood in the elderly. We sought to identify risk factors associated with morbidity and mortality in geriatric patients by anticoagulation status undergoing hip fracture repair. **Methods:** Patients aged ≥65 years undergoing surgery for hip fracture repair were included (2009–2019) from a US-based hip fracture registry. Factors associated with 90-day mortality were determined using multivariable logistic regression and stratified by antithrombotic agent medication use prior to surgery. Direct oral anticoagulation (DOAC) medications were the largest group, and all antithrombotic agents were included in the delineation. **Results:** A total of 35,463 patients were identified, and 87.1% (*N* = 30,902) were DOAC-naïve. Risk factors for 90-day mortality in DOAC-naïve patients were an American Society of Anesthesiologist’s (ASA) classification ≥3 (odds ratio [OR] = 2.56, 95% confidence interval [CI] = 2.24–2.93), preoperative myocardial infarction (OR = 1.87, 95% CI = 1.33–2.64), male gender (OR = 1.73, 95% CI = 1.59–1.88), congestive heart failure (CHF) (OR = 1.64, 95% CI = 1.50–1.80), psychoses (OR = 1.27, 95% CI = 1.15–1.42), renal failure (OR = 1.29, 95% CI = 1.19–1.40), smoking history (OR = 1.19, 95% CI = 1.09–1.29), chronic pulmonary disease (OR = 1.14, 95% CI = 1.05–1.25), increasing age (OR = 1.07, 95% CI = 1.06–1.07), and decreasing body mass index (BMI) (OR = 1.06, 95% CI = 1.05–1.08). Identified factors for mortality in the DOAC group also included ASA classification ≥3 (OR = 2.15, 95% CI = 1.44–3.20), male gender (OR = 1.68, 95% CI = 1.41–2.01), CHF (OR = 1.45, 95% CI = 1.22–1.73), chronic pulmonary disease (OR = 1.34, 95% CI = 1.12–1.61), decreasing BMI (OR = 1.04, 95% CI = 1.02–1.06), and increasing age (OR = 1.02, 95% CI = 1.01–1.03). **Conclusions:** Regardless of preoperative DOAC status, ASA classification, gender, CHF, chronic pulmonary disease, lower BMI, and higher age are associated with an increased risk of mortality. Some of these comorbidities can be utilized for risk stratification prior to surgery.

## 1. Introduction

In orthopedic practice, hip fractures are amongst the most commonly encountered fractures, with over 250,000 occurring in the United States (US) annually [1]. The incidence of elderly hip fracture is expected to increase along with increases in life expectancy. Surgical repair is effective and the most common treatment following hip fracture; however, hip fracture mortality ranges from 8% to 36% [2]. This is projected to continue and is unlikely to improve anytime soon [3,4,5,6]. As surgical complexity and patient age, fragility, and comorbidities rise, optimizing surgical care can lead to better patient outcomes. Additionally, data driven discussion and decisions will be imperative in improving patient mortality.

Updated American Academy of Orthopaedic Surgeons (AAOS) clinical guidelines recommend surgery within 24 to 48 h, recognizing differences in resources available to support surgical care at various facilities [7]. Operative delay may be caused by medical-related reasons, such as stabilization of the patient’s preoperative medical condition, and also administrative-related reasons, such as operating room and surgeon availability. Preoperative wait time and medical optimization can vary amongst types of surgical teams and facility resources. Patient comorbidities, patient age, and polypharmacy can further complicate hip fracture management preoperatively [8,9,10]. A meta-analysis published in 2018 found that in addition to time to surgery (>2 days/2 days), malignancy, nursing home residence, pulmonary disease, diabetes, and cardiovascular disease significantly increased the risk of mortality after hip fracture [11].

Patients taking direct oral anticoagulants (DOACs) have been found to have significant delays in time to emergency hip fracture surgery so as to reverse the effects of these medications, which can also lead to poorer outcomes compared to those not anticoagulated [12]. DOAC reversal is the most common reason for prolonged preoperative wait times [13,14]. As of 2019, 11.5% of Medicare beneficiaries were on DOACs [15].

Therefore, using data from a US-based integrated healthcare system’s hip fracture registry, we sought to identify patient and perioperative risk factors for 30-day, 90-day, and 1-year mortality, as well as postoperative complications, while accounting for time to surgery. To tease out the differential effect for those patients delayed for the optimization of acute medical conditions, risk factors for patients who were DOAC-naïve and those who were using DOACs at the time of hip fracture were considered separately. Among patients undergoing hip fracture repair, we believe those on DOAC therapy will have higher 30, 90, and 1-year mortality as well as worse short-term morbidity compared to patients not on anticoagulation therapy.

## 2. Materials and Methods

### 2.1. Study Design and Data Source

We conducted a retrospective cohort study using a longitudinally maintained US integrated healthcare system’s hip fracture registry. This registry includes all surgically treated hip fractures performed within the integrated healthcare system, which covers over 12.6 million members in eight geographical regions. The hip fracture registry’s target population are patients with fractures of the femoral neck, intertrochanteric region, or subtrochanteric region, comprising nearly all operative, low-energy, fragility-type fractures in the elderly population. The Hip Fracture Registry identifies relevant hip fracture cases using International Classification of Diseases, Ninth Revision, Clinical Modification (ICD-9-CM, ICD-10-CM) diagnostic and procedure codes recorded into KP’s electronic medical record [EMR] and administrative claims. Diagnostic and procedure codes include those mandated by the National Healthcare Safety Network (NHSN) (ICD-9-CM: 79.XX, 81.52). Data are extracted electronically on a quarterly schedule and sent to a data repository for data management, validation, and reporting [16]. In brief, this registry is a surveillance tool for surgically treated hip fractures, including information on demographics, medical comorbidities, intraoperative details, implant information, and outcomes using the integrated electronic health record (EHR), administrative databases, and other institutional databases within the integrated healthcare system. The Hip Fracture Registry monitors 10 outcomes, including pneumonia, myocardial infarction, thromboembolic events (deep vein thrombosis [DVT] and pulmonary embolism [PE]), and mortality. Except for length of stay and mortality, outcomes are captured using ICD-9-CM diagnosis and procedure codes recorded into the EMR and administrative claims, which are adjudicated by clinical experts who review patient charts.

#### Inclusion and Exclusion Criteria

Patients aged ≥65 years who underwent surgery for a hip fracture between 2009 and 2019 in Southern California, Northern California, the Pacific Northwest, and Hawaii were included. The study was restricted to four of the eight regions in the healthcare system where care is provided in institutionally owned hospitals. These four regions represent over 10.3 of the 12.6 million members served. Procedures performed after 2019 were not included due to COVID-19 shelter-in-place policies that could influence surgical wait time and outcomes differently. Only repairs utilizing nail fixation, pinning fixation, sliding screw fixation, or hemiarthroplasty were included. Patients with pathological fractures, multiple fractures, open fractures (i.e., non-low energy fall or injured from motor vehicle accident), procedures in conjugation with other surgeries during the same hospital stay, or prior hip surgeries on the same side were excluded (Figure 1). Finally, patients who had a hospital transfer (3% of patients) and those missing surgical wait time (defined below 0.8% of patients) were excluded. In sum, 35,463 patients comprised the final study sample. Procedures were performed by 546 surgeons at 35 hospitals.

### 2.2. Outcomes of Interest

The primary outcome was mortality at 30 days, 90 days, and 1 year. Mortality information was obtained from the Social Security Administration (SSA), thus capturing all patients regardless of their insurance at the time of death.

Secondary outcomes were 30-day pneumonia, 90-day venous thromboembolism (VTE), and 90-day acute myocardial infraction (MI). Pneumonia, VTE (including deep venous thrombosis [DVT] and pulmonary embolism [PE]), and myocardial infarction (MI) were defined according to the Agency for Healthcare Research and Quality (AHRQ) quality indicators [17]. DVT and PE were manually validated by clinical content experts, including confirmation via ultrasound and/or computed tomography reports.

### 2.3. Risk Factors

Patient factors considered included age (per 1-year increment), body mass index (BMI) (per 1 kg/m^2^ increment), sex (men vs. women), smoking status (current, previous, vs. never), American Society of Anesthesiologists (ASA) classification (≥3 vs. 1–2), 90-day preoperative acute MI (yes vs. no), and diabetes (yes vs. no). Elixhauser medical comorbidities included (yes vs. no) chronic pulmonary disease, congestive heart failure, hypertension, psychoses, and renal failure [18]. Surgical factors considered included anesthesia type (neuraxial, conversion to general, other techniques, vs. general), surgery type (pinning, sliding screw, hemiarthroplasty, vs. nail), and hospital volume (low [0 to 124 cases per year], medium [125 to 186 cases per year], vs. high [≥187 cases per year]).

### 2.4. Covariates

Covariates included in regression analyses included region (Southern California, Pacific Northwest, Hawaii vs. Northern California) and yearly trend (operative year per 1-year increment) [19]. Due to the effect of preoperative wait time on outcome and local variation, this was also included as a covariate in regression analysis. Surgical wait time is defined as hours and minutes elapsed from either emergency department or inpatient admission to the surgical start time. The surgical start time is defined as a timestamp of the patient’s arrival in the operating room, the procedure start time, or incision time, whatever came first. All timestamps were documented in the EHR.

### 2.5. Preoperative Antithrombotic Agent Medication Use

Patients were stratified by preoperative antithrombotic agents’ medication status (yes/no). We identified anticoagulation medications dispensed in our integrated system within 100 days before surgery, including: aspirin, apixaban, betrixaban, edoxaban, fondaparinux, rivaroxaban, danaparoid, heparin, dalteparin, enoxaparin, tinzaparin, bivalirudin, dabigatran, desirudin, lepirudin, warfarin, abciximab, eptifibatide, tirofiban, cangrelor, caplacizumab, cilostazol, clopidogrel, dipyridamole, prasugrel, ticagrelor, ticlopidine, and vorapaxar. As DOACs were the largest class of medication assessed in our cohort, we will use this acronym to refer to the full antithrombotic agent class.

### 2.6. Statistical Analysis

The cohort was described using means, standard deviations, frequencies, and proportions. Crude incidence for outcomes was calculated as one minus the Kaplan–Meier estimator.

Multiple logistic regression models were used to identify factors associated with the outcomes of interest. All factors specified were included in regression models. Models included the covariates specified above with a smooth curve with internal knots at 5, 27.5, 50, 72.5, and 95th percentile using cubic spline to adjust for non-linear surgical wait time trends. Missing values were managed using mean imputation in the following covariates: ASA classification (6.0%), BMI (0.6%), smoking status (1.8%), medical comorbidities (<0.1%), an anesthesia type (0.5%). Odds ratios (OR) 95% and confidence intervals (CI) are presented [20]. Analyses were performed using R version 3.6.2. *p* < 0.05 was the statistical significance threshold used for this study and all tests were 2-sided.

## 3. Results

A total of 4561 (12.9%) of patients used DOACs prior to hip fracture repair, while 30,902 (87.1%) were DOAC-naïve. Characteristics of patients by DOAC use are presented in Table 1. Median patient age for patients with and without DOAC use was 83 and 84 years, respectively. Median preoperative wait time for DOAC-naïve patients was 20.3 h (interquartile range [IQR] = 13–27 h) compared to 26.2 h (IQR = 18–44 h) in the DOAC patient group.

### 3.1. Risk Factors in Preoperative DOAC-Naïve Patients

The crude 30-day, 90-day, and 1-year mortality rates for patients who were DOAC-naïve were 5.1%, 10.7%, and 20.8%, respectively (Table 2). Factors were similar across all time points of interest and included an ASA classification ≥3 compared to <3, a history of acute MI within 90 days preoperatively, men to women, congestive heart failure, nail fixation compared to pinning, renal failure, psychoses, general compared to neuraxial anesthesia, prior smoking compared to never smoking, chronic pulmonary disease, increasing age, and decreasing BMI. Diabetes was a risk factor for 1-year mortality only (OR = 1.08, 95% CI = 1.01–1.16). Hypertension was associated with a lower likelihood of 90-day mortality (OR = 0.90, 95% CI = 0.81–0.99).

Multivariable results and risk factors for secondary outcomes in DOAC-naïve patients are reported in Table 3. Risk factors for 30-day pneumonia in patients who used DOACs included chronic pulmonary disease, other anesthesia techniques compared to general anesthesia, ASA classification ≥ 3, male gender, nail fixation compared to pinning, congestive heart failure, prior and current smoking compared to never smoking, psychoses, an annual hospital case volume of 125–186 compared to ≥187 procedures, increasing age, and decreasing BMI; nail fixation compared to pinning, increasing BMI, and increasing age associated with likelihood of 90-day VTE; diabetes associated with a lower likelihood of VTE; history of acute MI within 90 days preoperatively, congestive heart failure, current smoking, ASA classification ≥ 3, diabetes, an annual hospital case volume of ≤124 compared to ≥187 procedures, male gender, renal failure, nail fixation compared to hemiarthroplasty, and increasing age associated with 90-day MI.

### 3.2. Risk Factors in Patients with Preoperative DOAC Use

The crude 30-day, 90-day, and 1-year mortality rates for patients with preoperative DOAC use were 8.3%, 16.3%, and 30.5%, respectively (Table 2). Similarly to DOAC-naïve patients, risk factors for mortality identified at all time points included an ASA classification ≥3 compared to <3, nail fixation compared to pinning, men to women, congestive heart failure, chronic pulmonary disease, increasing age, and decreasing BMI (Table 3). Nail pinning compared to hemiarthroplasty was a newly identified risk factor in patients who used DOACs only. A history of acute MI within 90 days preoperatively was associated with 30-day mortality only (OR = 1.60, 95% CI = 1.08–2.36), while a history of renal failure (OR = 1.28, 95% CI = 1.11–1.48) and prior smoking (OR = 1.17, 95% CI = 1.01–1.36) were associated with 1-year mortality only.

Table 3 presents multivariable results and risk factors for secondary outcomes in patients who used DOACs preoperatively. Risk factors for 30-day pneumonia in patients who used DOACs included chronic pulmonary disease, male gender, congestive heart failure, neuraxial compared to general anesthesia, decreasing BMI, and increasing age. A history of hypertension was associated with a lower likelihood of 30-day pneumonia. No factors were associated with likelihood of 90-day VTE. A history of acute MI within 90 days preoperatively, ASA classification ≥ 3, nail fixation compared to hemiarthroplasty, male gender, decreasing BMI, and increasing age were associated with 90-day MI.

## 4. Discussion

Hip fractures and repair have a staggeringly high mortality rate, with a 1-year rate of 27.3% in 2011 and 30% in 2019 [21,22]. Anticoagulant reversal is the most common rate-limiting factor that leads to surgical delay [13,14], and therefore we sought to determine how DOAC status modifies identification of other risk factors for poor outcomes beyond time to surgery in patients undergoing hip fracture repair. For healthcare teams, determining which comorbidities to prioritize while limiting delay is difficult. Based on our findings, more evidence-based decisions can be made. Our retrospective cohort study of hip fracture repairs found that the most consistently identified risk factors for mortality, pneumonia, and MI that could be used for efficient risk stratification included increasing age, decreasing BMI, male gender, a higher ASA classification, and a history of chronic pulmonary disease or congestive heart failure.

Multiple studies have determined the risk of hip fracture based on comorbidities [23,24,25], but few have reported the outcomes of repair based on comorbidities and anticoagulation status. Age was relevant regardless of DOAC status for mortality, in alignment with previous studies [23]. Similarly to previous evidence, non-DOAC group patients who were men, had an ASA classification greater than 2, a history of smoking [24], CHF, psychoses [25,26], renal failure [27], or preoperative MI had an increased risk of postoperative mortality [28], pneumonia [29], and MI. We determined that gender, ASA classification, and CHF were also risk factors for DOAC patients.

We found lower likelihood of mortality and postoperative morbidity outside of VTE with increases in BMI [30], as improved nutritional status has survival benefit. Interestingly, diabetes had a non-consequential effect on mortality other than 1-year mortality in the DOAC-naïve group. This might be related to the protective mechanism of an increased BMI; however, it did result in higher 90-day postoperative MI in the DOAC-naïve group. Further research must be conducted to determine if the identified risk factors in the DOAC-naïve group are related to premature surgical intervention or other logistic mechanisms, while the protective factors in the DOAC group may be secondary to surgical delay, which may give time for optimization. Understanding the increased risks regardless of anticoagulation status is particularly helpful in risk stratification discussions and decision-making.

As expected, pinning fixation patients had a much lower risk of mortality, VTE, and pneumonia as the underlying physical damage and length of surgery are much less for pinning fixation. In parallel, hemiarthroplasties demonstrated lower mortality and significantly lower 90-day MI in both AC groups, as Harris hip scores are lower and have decreased major reoperation rates [31].

In the DOAC group, mortality, pneumonia, and MI were more prevalent, due to cardiovascular disease, delay in surgical repair and mobilization; increased likelihood of general anesthesia, and perioperative bleeding [22]. Similarly, patients in the DOAC group had a decreased thrombosis rate, potentially due to anticoagulant effects. The largest risk factor for postoperative MI was preoperative MI, regardless of DOAC status. Postoperative MI risk is known to be higher in hip fracture repair compared to other orthopedic surgeries [32].

We found that a neuraxial approach in the non-DOAC group lowered mortality outcomes, in accordance with previous studies [33,34]. The benefits of a neuraxial blockade in comparison with general anesthesia may confer multifactorial benefits, including altered coagulation, increased blood flow, improvements in pain free breathing, and critically a reduction in surgical stress responses [35]. There was an increased likelihood of pneumonia in the DOAC group who received neuraxial anesthesia. Pneumonia in the hip fracture population has been a point of interest for some time. A systematic review in 2010 and a meta-analysis in 2014 demonstrated decreased mortality when neuraxial anesthesia was compared to general [36,37]. In other studies, it showed little difference [38]. These conflicting findings may in part be due to surgical delay practice changes. Surgical delay can necessitate increased pain control therapy and increase the evolution of pneumonia. Our findings provide evidence for anesthetic approach differences and/or whether it would be beneficial to wait for reversal or correction.

Higher hospital case volume has been theorized to be associated with decreased operating times, improved approaches, reduced surgical manipulation, and care teams in larger-volume hospitals with experience in preoperative and postoperative management. Similarly to a 2019 systematic review, we found little difference in postoperative outcomes based on hospital volume, other than low-volume hospitals having a higher risk of postoperative MI [39], likely due to decreased mobilization support and robust support staff. Additional research is necessary to assess these explanations.

### Limitations

There are many strengths of our study. First, the analysis investigated risk factors for outcomes after stratifying patients based on preoperative DOAC use. As such, the present study provides evidence-based findings for surgeons, anesthesiologists, and care teams that can be leveraged for patient risk stratification and decision-making based on preoperative DOAC status. Second, our large study sample permitted the evaluation of many patient and surgical factors that were prospectively collected into a validated hip fracture registry. Our population may be generalizable to a sizable portion of the US, as the integrated healthcare system comprises many socioeconomic regions and locations, with the only limitation being that the patients have insurance coverage through employment, Medicare-type systems, or our national government-assisted insurance program [16]. Our patient sample size also enabled the investigation of multiple independent risk factors for mortality, including several important Elixhauser comorbidities by DOAC status. Importantly, we discovered that regardless of DOAC status, there were many overlapping risk factors between groups. Future research should build upon this finding and focus on risk factors that can be targeted or modified prior to surgery.

Limitations should also be considered. Importantly, the present study was observational and exploratory in nature. Only associations are reported and not causation. Despite our analytic techniques, there remains the possibility of residual confounding (incomplete controlling). Additional study in other cohorts is needed to confirm novel risk factors identified. Further, we were limited to factors collected in our healthcare system’s hip fracture registry. Additional factors of clinical relevance could not be evaluated as they were not captured by the registry. These factors included exact time of the fracture, surgeon’s years in training, use of antifibrinolytics, distance traveled, and other hospital factors outside of repair volume. While the large number of medical centers and operating surgeons may increase the generalizability of our study, we did not assess the standards of surgical techniques or related complications that could lead to delayed healing and increased morbidity and mortality [40,41]. Future studies must be conducted in order to determine whether optimization of the modifiable factors identified mitigates risk for outcomes. Finally, as our analysis stratified patients by preoperative DOAC use, we did not evaluate the interplay between specific DOACs on patient or surgical risk factors.

## 5. Conclusions

As expected, 30-day, 90-day, and 1-year mortality rates following hip fracture repair were higher in patients using DOAC. Our cohort study of more than 35,000 hip fracture repair patients over an 11-year period identified several risk factors for adverse outcomes following geriatric hip fracture repair. Regardless of preoperative DOAC status, ASA classification, gender, CHF, chronic pulmonary disease, lower BMI, and higher age are associated with increased risk of mortality. Some of these comorbidities can be utilized for risk stratification prior to surgery.

## Figures and Tables

**Figure 1 geriatrics-10-00054-f001:**
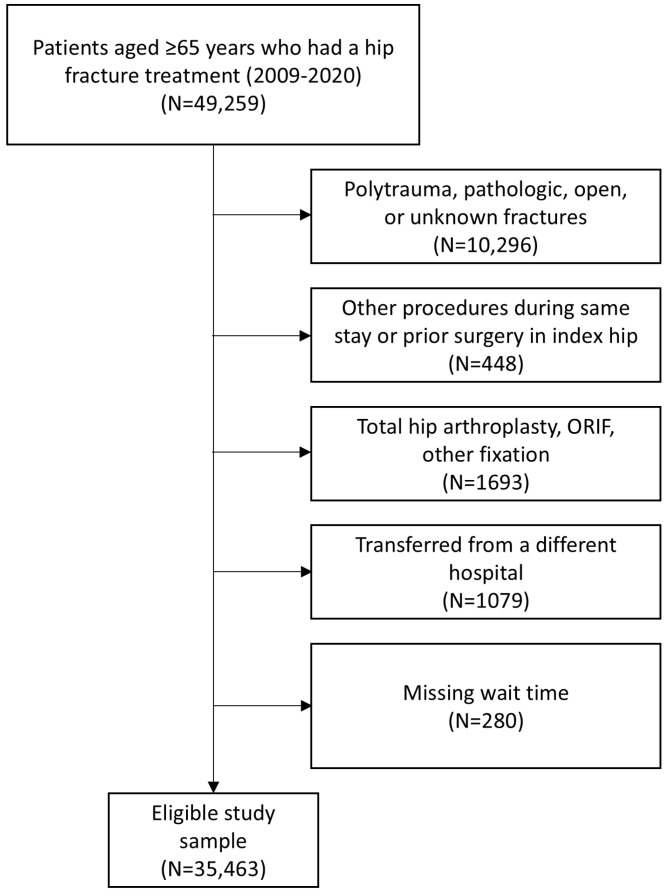
Patient inclusion and exclusion flowchart.

**Table 1 geriatrics-10-00054-t001:** Characteristics of 35,463 patients who underwent hip fracture repair in a US integrated healthcare system (2009–2019) by preoperative direct oral anticoagulation (DOAC) medication use.

Characteristic, *n* (%) Unless Specified	DOAC-Naïve	DOAC Use
Total N	30,902	4561
Patient factors		
Age, in years, mean (SD)	82.5 (8.1)	82.8 (7.7)
Median (IQR)	83 (77–89)	84 (78–88)
BMI, kg/m^2^, mean (SD)	23.8 (4.8)	24.3 (4.8)
Median (IQR)	23.3 (20.6–26.5)	23.7 (21.0–27.0)
Men	9092 (29.4)	1620 (35.5)
Smoking status		
Never	16,264 (53.6)	2215 (49.0)
Previous smoker	12,110 (39.9)	2060 (45.6)
Current smoker	1946 (6.4)	246 (5.4)
ASA classification ≥ 3	21,551 (74.0)	3794 (89.7)
90-day preoperative acute MI	191 (0.6)	251 (5.5)
Diabetes	8583 (27.8)	1599 (35.1)
Elixhauser medical comorbidities		
Chronic pulmonary disease	7718 (25.0)	1458 (32.0)
Congestive heart failure	5032 (16.3)	1640 (36.0)
Hypertension	23,941 (77.5)	3939 (86.4)
Psychoses	4280 (13.9)	827 (18.1)
Renal failure	9965 (32.3)	1921 (42.1)
Surgical factors		
Anesthesia type		
General	16,734 (54.4)	2895 (63.9)
Neuraxial	13,214 (42.9)	1509 (33.3)
Converted to general	473 (1.5)	59 (1.3)
Monitored anesthesia care	216 (0.7)	48 (1.1)
Regional	132 (0.4)	23 (0.5)
Procedure		
Nail fixation	12,679 (41.0)	1776 (38.9)
Pinning fixation	3825 (12.4)	622 (13.6)
Sliding screw fixation	2392 (7.7)	347 (7.6)
Hemiarthroplasty	12,006 (38.9)	1816 (39.8)
Hospital volume, in cases/year		
Low ≤ 125	8365 (27.1)	1300 (28.5)
Medium 126–186	8892 (28.8)	1309 (28.7)
High ≥187	13,645 (44.2)	1952 (42.8)
Surgical wait time, in hours, mean (SD)	23.8 (20.0)	35.7 (33.3)
Percentile: 5%, 25%, 50%, 75%, 95%	5.1, 13.2, 20.3, 27.4, 53.3	6.9, 17.9, 26.2, 44.3, 91.7

ASA = American Society of Anesthesiologists. BMI = body mass index. IQR = interquartile range. SD = standard deviation. Missing: ASA classification = 2114 (6.0%), BMI = 201 (0.6%), smoking status = 622 (1.8%), Elixhauser comorbidities = 8 (<0.1%), anesthesia type = 160 (0.5%).

**Table 2 geriatrics-10-00054-t002:** Risk factors for mortality for patients who underwent hip fracture repair stratified by patients with and without preoperative direct oral anticoagulation medications and adjusted for with surgical wait times.

Risk Factors	30-Day Mortality		90-Day Mortality		1-Year Mortality	
	DOAC-Naïve	DOAC Use	DOAC-Naïve	DOAC Use	DOAC-Naïve	DOAC Use
	OR (95% CI)	OR (95% CI)	OR (95% CI)	OR (95% CI)	OR (95% CI)	OR (95% CI)
Age: per 1 incr.	**1.06 (1.05–1.07)**	**1.05 (1.04–1.07)**	**1.07 (1.06–1.07)**	**1.06 (1.05–1.07)**	**1.06 (1.06–1.06)**	**1.06 (1.05–1.07)**
Men vs. women	**1.90 (1.70–2.12)**	**1.63 (1.29–2.06)**	**1.73 (1.59–1.88)**	**1.68 (1.41–2.01)**	**1.74 (1.63–1.86)**	**1.75 (1.51–2.02)**
ASA: 3+ vs. 1–2	**2.94 (2.40–3.61)**	**1.74 (1.02–2.98)**	**2.56 (2.24–2.93)**	**2.15 (1.44–3.20)**	**2.55 (2.32–2.80)**	**2.27 (1.68–3.07)**
BMI: per 1 incr.	**0.94 (0.93–0.95)**	**0.97 (0.94–0.99)**	**0.94 (0.93–0.95)**	**0.96 (0.94–0.98)**	**0.93 (0.93–0.94)**	**0.94 (0.92–0.95)**
Smoking status: ref. never-smokers						
Current smokers	0.84 (0.64–1.09)	1.65 (1.00–2.72)	0.83 (0.68–1.01)	1.21 (0.81–1.81)	1.05 (0.92–1.21)	1.35 (0.98–1.86)
Previous smokers	**1.16 (1.03–1.30)**	1.26 (0.99–1.61)	**1.19 (1.09–1.29)**	1.15 (0.96–1.38)	**1.22 (1.14–1.30)**	**1.17 (1.01–1.36)**
Diabetes	0.93 (0.82–1.05)	0.94 (0.74–1.21)	0.98 (0.90–1.08)	1.00 (0.83–1.20)	**1.08 (1.01–1.16)**	1.06 (0.91–1.24)
Chronic pulmonary disease	**1.17 (1.04–1.32)**	**1.27 (1.00–1.62)**	**1.14 (1.05–1.25)**	**1.34 (1.12–1.61)**	**1.17 (1.09–1.25)**	**1.40 (1.20–1.62)**
Congestive heart failure	**1.64 (1.45–1.85)**	**1.64 (1.30–2.06)**	**1.64 (1.50–1.80)**	**1.45 (1.22–1.73)**	**1.76 (1.63–1.89)**	**1.55 (1.35–1.79)**
90-day preoperative MI	**2.38 (1.59–3.57)**	**1.60 (1.08–2.36)**	**1.87 (1.33–2.64)**	1.37 (1.00–1.88)	**1.67 (1.22–2.27)**	1.21 (0.92–1.60)
Hypertension	0.89 (0.78–1.03)	0.75 (0.54–1.03)	**0.90 (0.81–0.99)**	0.83 (0.65–1.06)	0.96 (0.89–1.04)	1.03 (0.83–1.27)
Psychoses	**1.25 (1.08–1.44)**	1.02 (0.77–1.36)	**1.27 (1.15–1.42)**	0.99 (0.80–1.23)	**1.24 (1.14–1.35)**	0.99 (0.83–1.18)
Renal failure	**1.24 (1.11–1.39)**	1.19 (0.95–1.50)	**1.29 (1.19–1.40)**	1.16 (0.97–1.37)	**1.27 (1.19–1.35)**	**1.28 (1.11–1.48)**
Anesthesia: ref. general						
Neuraxial	**0.80 (0.71–0.89)**	0.94 (0.74–1.19)	**0.82 (0.76–0.89)**	0.95 (0.80–1.14)	**0.85 (0.80–0.90)**	0.94 (0.81–1.08)
Converted to general	1.15 (0.78–1.69)	1.25 (0.49–3.24)	1.06 (0.79–1.42)	1.24 (0.61–2.53)	1.00 (0.79–1.27)	0.98 (0.53–1.80)
Others	1.26 (0.82–1.94)	0.98 (0.38–2.55)	1.29 (0.94–1.78)	1.14 (0.60–2.17)	1.05 (0.80–1.36)	1.41 (0.84–2.37)
Procedure: ref. nail						
Pinning	**0.57 (0.47–0.71)**	**0.43 (0.28–0.66)**	**0.65 (0.57–0.75)**	**0.59 (0.45–0.79)**	**0.78 (0.71–0.87)**	**0.77 (0.62–0.96)**
Sliding screw	1.00 (0.82–1.21)	0.77 (0.50–1.19)	1.02 (0.88–1.17)	0.72 (0.51–1.00)	0.98 (0.87–1.09)	0.77 (0.59–1.01)
Hemi	0.89 (0.80–1.00)	**0.77 (0.61–0.98)**	0.93 (0.86–1.01)	**0.78 (0.65–0.94)**	0.95 (0.89–1.02)	**0.82 (0.70–0.95)**
Hospital volume: ref. high, ≥187 cases/year						
Low, ≤124 cases/year	1.07 (0.93–1.22)	0.81 (0.61–1.07)	1.04 (0.94–1.15)	0.97 (0.78–1.20)	1.07 (1.00–1.16)	1.05 (0.88–1.24)
Medium, 125–286 cases/year	1.07 (0.95–1.22)	0.87 (0.66–1.13)	1.05 (0.96–1.15)	0.96 (0.79–1.18)	1.05 (0.98–1.13)	0.98 (0.83–1.15)

No outcomes, OR cannot be estimated. DOAC = direct oral anticoagulation medication. OR = odds ratio. CI = confidence interval. VTE = venous thromboembolism. MI = myocardial infraction. Bold indicates significance *p* < 0.05.

**Table 3 geriatrics-10-00054-t003:** Risk factors for pneumonia, VTE, and MI for patients who underwent hip fracture repair stratified by patients with and without preoperative anticoagulation medications and adjusted for with surgical wait times.

Risk Factors	30-Day Pneumonia		90-Day VTE		90-Day MI	
	DOAC-Naïve	DOAC Use	DOAC-Naïve	DOAC Use	DOAC-Naïve	DOAC Use
	OR (95% CI)	OR (95% CI)	OR (95% CI)	OR (95% CI)	OR (95% CI)	OR (95% CI)
Age: per 1 incr.	**1.04 (1.03–1.04)**	**1.02 (1.01–1.03)**	**1.02 (1.01–1.03)**	1.00 (0.97–1.04)	**1.03 (1.01–1.04)**	**1.03 (1.00–1.06)**
Men vs. women	**1.56 (1.43–1.71)**	**1.48 (1.21–1.81)**	0.92 (0.78–1.08)	0.74 (0.46–1.19)	**1.30 (1.06–1.59)**	**1.79 (1.23–2.61)**
ASA: 3+ vs. 1–2	**1.60 (1.41–1.82)**	1.49 (0.98–2.27)	0.88 (0.74–1.05)	0.87 (0.43–1.75)	**1.49 (1.10–2.03)**	**5.44 (1.35–21.92)**
BMI: per 1 incr.	**0.98 (0.97–0.99)**	**0.97 (0.95–1.00)**	**1.03 (1.02–1.05)**	1.03 (0.99–1.08)	0.99 (0.97–1.01)	**0.95 (0.91–1.00)**
Smoking status: ref. never smokers						
Current smokers	**1.22 (1.02–1.46)**	1.30 (0.84–2.00)	1.09 (0.78–1.52)	1.34 (0.52–3.47)	**1.63 (1.09–2.42)**	1.93 (0.92–4.06)
Previous smokers	**1.23 (1.12–1.34)**	1.21 (0.98–1.50)	1.16 (0.99–1.36)	1.34 (0.83–2.15)	1.21 (0.99–1.49)	1.11 (0.75–1.65)
Diabetes	1.03 (0.94–1.14)	1.10 (0.89–1.37)	**0.81 (0.68–0.97)**	0.98 (0.59–1.60)	**1.47 (1.20–1.81)**	1.28 (0.87–1.88)
Chronic pulmonary disease	**1.86 (1.70–2.04)**	**1.78 (1.45–2.18)**	1.07 (0.90–1.28)	1.38 (0.87–2.21)	0.90 (0.72–1.12)	0.94 (0.64–1.40)
Congestive heart failure	**1.34 (1.21–1.49)**	**1.33 (1.08–1.63)**	0.92 (0.75–1.14)	1.07 (0.66–1.74)	**1.97 (1.60–2.43)**	1.32 (0.91–1.92)
90-day preoperative AMI	1.21 (0.80–1.83)	1.11 (0.76–1.62)	1.30 (0.57–2.96)	1.43 (0.60–3.40)	**55.64 (40.32–76.78)**	**24.71 (17.01–35.91)**
Hypertension	1.03 (0.92–1.15)	**0.68 (0.51–0.89)**	0.95 (0.79–1.13)	0.58 (0.33–1.03)	1.21 (0.91–1.61)	1.46 (0.76–2.79)
Psychoses	**1.13 (1.00–1.27)**	1.07 (0.83–1.37)	1.00 (0.81–1.24)	0.78 (0.43–1.43)	1.17 (0.90–1.52)	0.76 (0.46–1.24)
Renal failure	1.02 (0.93–1.12)	1.06 (0.87–1.31)	0.90 (0.76–1.07)	0.78 (0.49–1.26)	**1.29 (1.05–1.58)**	1.13 (0.79–1.63)
Anesthesia: ref. general						
Neuraxial	1.00 (0.91–1.09)	**1.25 (1.02–1.54)**	0.97 (0.83–1.13)	0.93 (0.58–1.51)	0.86 (0.70–1.05)	1.02 (0.68–1.51)
Converted to general	0.84 (0.59–1.21)	1.95 (0.93–4.09)	0.82 (0.43–1.55)	0.84 (0.11–6.33)	1.30 (0.67–2.54)	--*
Others	**0.55 (0.34–0.90)**	1.57 (0.81–3.04)	0.56 (0.21–1.52)	0.76 (0.10–5.75)	0.56 (0.21–1.46)	1.35 (0.56–3.26)
Procedure: ref. nail						
Pinning	**0.74 (0.63–0.86)**	0.83 (0.60–1.13)	**0.47 (0.35–0.64)**	0.56 (0.26–1.21)	0.82 (0.59–1.14)	1.18 (0.70–1.97)
Sliding screw	1.00 (0.85–1.17)	0.90 (0.62–1.32)	0.77 (0.57–1.04)	1.15 (0.54–2.44)	0.91 (0.63–1.32)	0.80 (0.37–1.72)
Hemi	1.00 (0.91–1.10)	0.83 (0.67–1.03)	1.00 (0.86–1.17)	0.78 (0.48–1.27)	**0.79 (0.64–0.98)**	**0.55 (0.37–0.83)**
Hospital volume: ref. high, ≥187 cases/year						
Low, ≤124 cases/year	1.09 (0.98–1.22)	1.14 (0.89–1.46)	0.97 (0.80–1.17)	1.38 (0.80–2.37)	**1.45 (1.14–1.85)**	0.75 (0.47–1.20)
Medium, 125–286 cases/year	**1.11 (1.01–1.23)**	1.19 (0.94–1.50)	0.86 (0.72–1.02)	0.84 (0.48–1.47)	1.24 (0.99–1.57)	0.79 (0.51–1.21)

* No outcomes, OR cannot be estimated. DOAC = direct oral anticoagulation medication. OR = odds ratio. CI = confidence interval. VTE = venous thromboembolism. MI = myocardial infarction. Bold indicates significance *p* < 0.05.

## Data Availability

Data used for this study are unavailable due to privacy or ethical restrictions.

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
