# Peer review of "Factors Associated with Mortality and Short-Term Patient Outcomes for Hip Fracture Repair in the Elderly Based on Preoperative Anticoagulation Status"

_geriatrics, 2025, doi:10.3390/geriatrics10020054_

Round 1

Reviewer 1 Report

Comments and Suggestions for Authors

Dear Authors,

I was pleased to review the paper entitled " Factors associated with mortality and short-term patient outcomes for hip fracture repair in the elderly based on preoperative anticoagulation status" -

- MDPI –

The present paper is very interesting, it focuses on a relevant clinical scenario, for orthopedics, potentially influencing the surgical and clinical practice for the management of hip fracture mortality.

Therefore, it is my opinion that the content is original, current, and relevant.

Thus, there are some minor remarks:

- Title: The title gives a fine idea of the topic to be covered.

- Abstract: please improve background seems to be redundant.

- Introduction: Conclude the introduction with your hypothesis on mortality in patients taking these drugs

- Method: Add more details in patient selection, such as which ICD-9-CM codes were used in the search.

- Results: correct

- Discussion: The assessment of these risks in patients using DOAC, should not exempt the reader from remembering the proper performance of surgery in these patients is crucial. Many of the complications also result from incorrect surgical technique that leads the patient to prolonged bed rest, and thus the onset of mortality (you could add from doi: 10.1302/2633-1462.56.BJO-2023-0163.R1).

You wrote “We found that neuraxial approach in the non-DOAC group lower mortality out-261 comes in accordance with previous studies”: According to your research what could be the reason? Learn more about it.

The limitations are very well written. Improve as clearly the strengths of this study and dwell more on the future implications of your research.

The paper generally is well written and needs only minor changes.

Author Response

Dear Reviewer,

Thank you very much for your kind comments and thorough review and for the opportunity to improve our manuscript. We have numbered the comments/questions and our responses are below in red.

Comment #1- Title: The title gives a fine idea of the topic to be covered.

Comment #1 Author Response: Thank you for your feedback.

Comment #2- Abstract: please improve background seems to be redundant.

Comment #2 Author Response: Thank you for your suggestion. We have added the following sentence:

(Line 13) “The one-year mortality risk for elderly patients undergoing proximal femur fracture repair surgery is 3-4 times higher compared to the general population”.

We have modified the background of the abstract to be the following:

(Lines 15-17): “Other than time to surgery, risk factors for postoperative morbidity and mortality following surgery are poorly understood in the elderly. We sought to identify risk factors associated with morbidity and mortality in geriatric patients by anticoagulation status undergoing hip fracture repair.”

Comment #3- Introduction: Conclude the introduction with your hypothesis on mortality in patients taking these drugs

Comment #3 Author Response: Thank you for your suggestion. We didn’t originally include a hypothesis statement because the purpose of our paper was to identify independent risk factors for morbidity and mortality in hip fracture repair patients. We agree that because we have separated patients by direct oral anticoagulation (DOAC) status, the paper would benefit from a hypothesis statement about the difference in mortality and short term outcomes between the two groups. We have added our hypothesis statement on for mortality for the two groups of patients identified by anticoagulation status.

(Lines 69-72): “Among patients undergoing hip fracture repair, we believe those on DOAC therapy will have higher 30, 90, and 1-year mortality as well as worse short-term morbidity compared to patients not on anticoagulation therapy”.

Comment #4- Method: Add more details in patient selection, such as which ICD-9-CM codes were used in the search.

Comment #4 Author Response: Thank you for your comment. The data used for this study is from an existing hip fracture registry at Kaiser Permanente. I have included a publication detailing the specifics of the hip fracture registry as well as a summary of the how the data for the hip fracture registry is captured.

(Lines 78-87)“The hip fracture registry’s target population are patients with fractures of the femoral neck, intertrochanteric region, or subtrochanteric region, comprising nearly all opera-tive, low-energy, fragility-type fractures in the elderly population. The Hip Fracture Registry identifies relevant hip fracture cases using International Classification of Dis-eases, Ninth Revision, Clinical Modification (ICD-9-CM, ICD-10-CM), diagnostic and procedure codes recorded into KP’s EMR and administrative claims. Diagnostic and procedure codes include those mandated by the National Healthcare Safety Network (NHSN) (ICD-9-CM: 79.XX, 81.52). Data are extracted electronically on a quarterly schedule and sent to a data repository for data management, validation, and reporting.”

We have added the following information regarding the capture of outcomes:

(Lines 91-96): “The Hip Fracture Registry monitors 10 outcomes including pneumonia, myocardial infarction, thromboembolic events (deep vein thrombosis [DVT] and pulmonary embolism [PE]), and mortality. Except for length of stay and mortality outcomes are captured using ICD-9-CM diagnosis and procedure codes recorded into the EMR and administrative claims which are adjudicated by clinical experts who review patient charts.”

 Comment #5- Results: correct

Comment #5 Author Response: Thank you

Comment #6-- Discussion: The assessment of these risks in patients using DOAC, should not exempt the reader from remembering the proper performance of surgery in these patients is crucial. Many of the complications also result from incorrect surgical technique that leads the patient to prolonged bed rest, and thus the onset of mortality (you could add from doi: 10.1302/2633-1462.56.BJO-2023-0163.R1).

Comment #6 Author Response: Thank you for your comment. We agree that while we have included many patient and surgical factors improper surgical technique can greatly delay patient healing and lead to poor outcomes. We have included this as a limitation in our discussion section as well as the above-mentioned article and another on surgical technique and complication.

(Lines 309-312): Finally, while the large number of medical centers and operating surgeons may increase the generalizability of our study, we did not quality of surgical technique or complications which could lead to delayed healing and increased morbidity and mortality.”

 Comment #7-You wrote “We found that neuraxial approach in the non-DOAC group lower mortality out-261 comes in accordance with previous studies”: According to your research what could be the reason? Learn more about it.

Comment #7 Author Response: Thank you for your comment. Based on the literature it appears that the benefits conferred by a neuraxial blockade in comparison with general anesthesia are multifactorial including altered coagulation, increased blood flow, improved ability to breathe free of pain, and importantly a reduction in surgical stress responses. We have included this in the discussion.

(Lines 274-277): “The benefits of a neuraxial blockade, in comparison with general anesthesia, may confer multifactorial benefits including altered coagulation, increased blood flow, improve-ments in pain free breathing, and critically a reduction in surgical stress responses [35]”.

Comment #8: The limitations are very well written. Improve as clearly the strengths of this study and dwell more on the future implications of your research.

Comment #8 Author Response: Thank you for your comment. We have improved the strengths of our study and added a sentence on the future implications of this research.

(Lines 295-298): “As such, the present study provides evidence-based findings for surgeons, anesthesiologists, and care teams which can be leveraged for patient risk stratification and decision making based on preoperative DOAC status.

(Lines 304-309):”Our patient sample size also enabled the investigation of multiple independent risk factors for mortality including several important Elixhauser comorbidities by DOAC status. Importantly we discovered that regardless of DOAC status there were many overlapping risk factors between groups. Future research should build upon this finding and focus on risk factors that can be targeted or modified prior to surgery.”

The paper generally is well written and needs only minor changes.

Reviewer 2 Report

Comments and Suggestions for Authors

The manuscript entitled “Factors associated with mortality and short-term patient outcomes for hip fracture repair in the elderly based on preoperative anticoagulation status” is very interesting and original. It is written in a clear and easy-to-read manner, which increases its value.

An aging population carries an increased risk of hip fractures, however, despite excellent surgical techniques, it is associated with not only perioperative but also long-term mortality. The search for risk factors associated with increased mortality is extremely valuable and has a large clinical aspect.My only comment concerns whether patients treated with DOACs before surgery were given drugs that reversed their action (antidote), e.g. idarucizumab, andexanet alfa. Do the authors believe that their use can shorten the time from injury to orthopedic surgery?

Author Response

Reviewer 2

The manuscript entitled “Factors associated with mortality and short-term patient outcomes for hip fracture repair in the elderly based on preoperative anticoagulation status” is very interesting and original. It is written in a clear and easy-to-read manner, which increases its value.

Comment #1: An aging population carries an increased risk of hip fractures, however, despite excellent surgical techniques, it is associated with not only perioperative but also long-term mortality. The search for risk factors associated with increased mortality is extremely valuable and has a large clinical aspect. My only comment concerns whether patients treated with DOACs before surgery were given drugs that reversed their action (antidote), e.g. idarucizumab, andexanet alfa. Do the authors believe that their use can shorten the time from injury to orthopedic surgery?

Comment #1 Author Response: Thank you so much for your thorough review and thoughtful comments. Unfortunately, we were unable to ascertain which DOAC patients received specific reversal agents or non-specific prohemostatic agents compared to the “time-reversal” strategy prior to hip fracture surgery. Based on a 2021 meta-analysis, patients with hip fracture on a DOAC who did not receive anticoagulant reversal had a shorter time to surgery (TTS) and shorter length of stay (LOS) compared to those who received anticoagulant reversal with no difference in blood loss or mortality rate although we could not assess this in our population.

References:

  1. Meinig R, Jarvis S, Orlando A, Nwafo N, Banerjee R, McNair P, Woods B, Harrison P, Nentwig M, Kelly M, Smith W, Bar-Or D. Is anticoagulation reversal necessary prior to surgical treatment of geriatric hip fractures? J Clin Orthop Trauma. 2020 Feb;11(Suppl 1):S93-S99. doi: 10.1016/j.jcot.2019.10.004. Epub 2019 Oct 15. PMID: 31992926; PMCID: PMC6977537.
  2. You D, Xu Y, Ponich B, et al. Effect of oral anticoagulant use on surgical delay and mortality in hip fracture. Bone Joint J. 2021;103-B(2):222-233. doi:10.1302/0301-620X.103B2.BJJ-2020-0583.R2

Reviewer 3 Report

Comments and Suggestions for Authors

Dear Authors, 

Hip surgery following hip fractures in the elderly and the mortality associated with it are an important topic specifically due to the rapid increase of geriatric population. Moreover, I consider the stratification based on the presence of antithrombotic agents useful, as these patients often receive delayed surgery due to possible complications. 

Below you may find my specific comments regarding your manuscript. 

General comment: You use the term DOAC (direct oral anticoagulants), which comprises explicitly the anti-factor-Xa and anti-factor-II anticoagulants namely rivaroxaban, apixaban, edoxaban, bivalrudin, enoxaparin etc. Aspirin, clopidogrel, prasugrel, ticagrelor etc. are antiplatelet drugs having a different indication and mechanism of action. Warfarin is also an anti-vitamin K anticoagulant and not a DOAC. Please consider the use of "antithrombotic agents" if you would like to include all the specified substances in your study. 

Abstract: 

Line 27: Please check the sentence, I think you refer to the patients using DOACs. 

Methods: 

Please specify why you excluded the total hip arthroplasties. 

Line 217: Figure 1? 

Author Response

Reviewer 3

Dear Authors,

Hip surgery following hip fractures in the elderly and the mortality associated with it are an important topic specifically due to the rapid increase of geriatric population. Moreover, I consider the stratification based on the presence of antithrombotic agents useful, as these patients often receive delayed surgery due to possible complications.

Below you may find my specific comments regarding your manuscript.

Comment #1 General comment: You use the term DOAC (direct oral anticoagulants), which comprises explicitly the anti-factor-Xa and anti-factor-II anticoagulants namely rivaroxaban, apixaban, edoxaban, bivalrudin, enoxaparin etc. Aspirin, clopidogrel, prasugrel, ticagrelor etc. are antiplatelet drugs having a different indication and mechanism of action. Warfarin is also an anti-vitamin K anticoagulant and not a DOAC. Please consider the use of "antithrombotic agents" if you would like to include all the specified substances in your study.

Comment #1 Author Response: Thank you very much for your thoughtful comments and thorough review. We have adopted your suggestion in the abstract and methodology as below.

(Lines 19-22) “Factors associated with 90-day mortality were determined using multivariable logistic regression and stratified by antithrombotic agent medication use prior to surgery. The direct oral anticoagulation (DOAC) medications were the largest group, and all antithrombotic agent were included in the delineation.”

(Lines 147-148) “Patients were stratified by preoperative antithrombotic agents’ medication status (yes/no).”

(Lines 153-154): “As DOACs were the largest class of medication assessed in our cohort we will use this acronym to refer to the full antithrombotic agent class.

Abstract:

Comment #2: Line 27: Please check the sentence, I think you refer to the patients using DOACs.

Comment #2 Author Response: Thank you we have corrected the error.

(Lines 29-33): “Identified factors for mortality in the DOAC group also included ASA ≥3 (OR=2.15, 95% CI=1.44-3.20), male gender (OR=1.68, 95% CI=1.41-2.01), CHF (OR=1.45, 95% CI=1.22-1.73), chronic pulmonary disease (OR=1.34, 95% CI=1.12-1.61), decreasing BMI (OR=1.04, 95% CI=1.02-1.06), and increasing age (OR=1.02, 95% CI=1.01-1.03)”.

Methods:

Comment #3: Please specify why you excluded the total hip arthroplasties.

Comment #3 Author Response: Total hip arthroplasties (THA) and Open Reduction and Internal Fixation (ORIF) comprised only 1,693 total procedures compared to the 35,463 procedures included in the cohort. They were removed to reduce heterogeneity across indications for procedure.

Comment #4: Line 217: Figure 1?

Comment #4 Author Response: This error has been corrected and the following sentence has been removed.

(Line 217) “Figure 1. This is a figure. Schemes follow the same formatting.”